# Does Iodine Intake Modify the Effect of Maternal Dysglycemia on Birth Weight in Mild-to-Moderate Iodine-Deficient Populations? A Mother–Newborn Prospective Cohort Study

**DOI:** 10.3390/nu15132914

**Published:** 2023-06-27

**Authors:** Yaniv S. Ovadia, Dov Gefel, Yoel Toledano, Shani R. Rosen, Yael Avrahami-Benyounes, Ludmila Groisman, Efrat Rorman, Lihi Hen, Shlomo Fytlovich, Liora S. Katz, Eyal Y. Anteby, Simon Shenhav

**Affiliations:** 1Obstetrics and Gynecology Department, Barzilai University Medical Center, Ashkelon 7830604, Israel; lihilililucas@yahoo.com (L.H.); eyala@bmc.gov.il (E.Y.A.); simon-64@zahav.net.il (S.S.); 2Foreign Studies Department, Robert H. Smith Faculty of Agriculture, Food and Environment, The Hebrew University of Jerusalem, Rehovot 76100001, Israel; 3School of Nutritional Science, Institute of Biochemistry, Food Science and Nutrition, Robert H. Smith Faculty of Agriculture, Food and Environment, The Hebrew University of Jerusalem, Rehovot 76100001, Israel; dubigefel@gmail.com (D.G.); shani.rosen1@mail.huji.ac.il (S.R.R.); 4Endocrinology Clinic, Division of Maternal Fetal Medicine, Helen Schneider Women’s Hospital, Rabin Medical Center, Petah Tikva 4941492, Israel; toledanoyoel@gmail.com; 5Center for Healthcare Technology and Innovation Policy Research, Gertner Institute of Epidemiology & Health Policy Research, Sheba Medical Center, Ramat Gan 5262000, Israel; 6Women’s Health Center, Maccabi Healthcare Services, Southern Region, Beersheba 8489312, Israel; yaelavrahami@gmail.com; 7National Public Health Laboratory, Ministry of Health, Tel Aviv 6108401, Israel; luda.groisman@phlta.health.gov.il (L.G.); efrat.rorman@phlta.health.gov.il (E.R.); 8Laboratory of Clinical Biochemistry, Barzilai University Medical Center, Ashkelon 7830604, Israel; shlomof@netvision.net.il; 9Diabetes, Obesity and Metabolism Institute, Icahn School of Medicine at Mount Sinai, New York, NY 10029, USA; liora.katz@mssm.edu; 10Faculty of Health Sciences, Ben-Gurion University of Negev, Beersheba 8410501, Israel

**Keywords:** iodine deficiency, glucose challenge test, large for gestational age, thyroglobulin, iodine-containing supplements, maternal dysglycemia

## Abstract

It is unclear how maternal glycemic status and maternal iodine status influence birth weight among individuals with mild-to-moderate iodine deficiency (ID). We studied the association between birth weight and both maternal glucose levels and iodine intake among pregnant women with mild-to-moderate ID. Glucose values were assessed using a glucose challenge test (GCT) and non-fasting glucose levels that were determined before delivery; individuals’ iodine statuses were assessed using an iodine food frequency questionnaire; and serum thyroglobulin (Tg) and urinary iodine concentrations (UIC) were used to assess each group’s iodine status. Thyroid antibodies and free thyroxine (FT4) levels were measured. Obstetric and anthropometric data were also collected. Large-for-gestational age (LGA) status was predicted using a Cox proportional hazards model with multiple confounders. Tg > 13 g/L was independently associated with LGA (adjusted hazard ratio = 3.4, 95% CI: 1.4–10.2, *p* = 0.001). Estimated iodine intake correlated with FT4 among participants who reported consuming iodine-containing supplements (ICS) after adjusting for confounders (β = 0.4, 95% CI: 0.0002–0.0008, *p* = 0.001). Newborn weight percentiles were inversely correlated with maternal FT4 values (β = −0.2 95% CI:−0.08–−56.49, *p* = 0.049). We conclude that in mild-to-moderate ID regions, insufficient maternal iodine status may increase LGA risk. Iodine status and ICS intake may modify the effect that maternal dysglycemia has on offspring weight.

## 1. Introduction

Birth weight is considered the most important marker of fetal growth and development in utero and reflects the adaptation of the fetus to the intrauterine environment. Small-for-gestational age (SGA) newborns have an increased risk of prenatal mortality and of suffering later in life from diseases such as metabolic syndrome, coronary heart disease, hypertension, stroke, kidney disease, osteoporosis, depression, and persistent anemia [1,2,3,4]. Large-for-gestational age (LGA) newborns have a greater risk of suffering from neonatal metabolic abnormalities, including hyperglycemia, birth trauma, stillbirth, and neonatal death [5]. Individuals with LGA might also be at an increased risk for overweight, obesity, cardiovascular disease, and diabetes later in life [5,6]. Gestational Diabetes Mellitus (GDM) is associated with an increased risk of developing LGA. Moreover, preliminary data have shown that even elevated maternal glucose levels that do not meet the diagnostic threshold for GDM are associated with increased birth weight [7]. Additionally, the results from a retrospective study including screening tests that used the 50 g 1 h post-oral glucose challenge test (GCT), which was performed at 24–28 weeks of pregnancy, suggest that the 50 g GCT can be used to identify women at risk of delivering offspring with an excessive delivery weight [8]. Importantly, the GCT is typically performed in the second half of pregnancy. However, interventions aiming to reduce the risk of fetal overgrowth and its associated health complications may not be effective if implemented at this late stage of pregnancy.

Iodine is critical for thyroid hormone synthesis, specifically triiodothyronine (T3) and thyroxine (T4). These hormones are important for normal human development, including the development of the fetal brain and nervous system, as well as cell metabolism [9]. Early in the first trimester, before the fetal thyroid gland is functioning, maternal thyroid hormones are transferred to the fetus as its sole source of thyroid hormones. Moreover, although the fetal thyroid gland is functionally mature around the gestational age of 18–20 weeks, the contribution of maternal thyroid hormones to fetal thyroid hormone availability is still considerable. Studies show a strong association between maternal thyroid function and fetal thyroid function [10,11]. Furthermore, studies show that a maximum of 30% to 50% of newborn T4 concentrations can be reached in the absence of fetal thyroid hormone production [12,13]. Thus, late-pregnancy maternal thyroid function may play an important role in fetal development. Essentially, in utero growth depends on adequate maternal iodine intake throughout pregnancy. Moreover, pregnant women are at high risk of developing iodine deficiency (ID) due to increased urinary iodine clearance and fetal iodine requirements [9].

We recently reported that inadequate maternal iodine intake may increase the risk of LGA among newborns, which was determined using thyroid function tests (*n* = 134) [14]. Following this, a study from China reported increased birthweight among newborns of mothers with ID (nearly significant [*p* = 0.07] *n* = 726) [15]. Along with thyroid function tests and thyroid antibodies, maternal iodine status was based on UIC in the latter study, while ours lacked consideration of UIC. Another limitation of our previous study was that maternal circulating glucose indices were not extensively explored. Hence, in the current study, we also used UIC measurements and examined maternal circulating glucose levels to assess the influence of elevated subclinical GDM glucose levels on birth weight [7]. Our current study aimed to explore the relationship between birth weight, glucose levels, and iodine intake among pregnant women with mild-to-moderate ID.

## 2. Materials and Methods

### 2.1. Design, Participants, Setting, and Ethics

This study was a longitudinal prospective cohort study; its methods have previously been described in detail [16]. In brief, the study was conducted during the period of June 2018 to April 2020. Pregnant women (*n* = 251) attending the Obstetrics and Gynecology Department of Barzilai University Medical Center, Ashkelon, Israel (BUMCA), were enrolled and screened for participation. Inclusion criteria were as follows: (1) planned delivery at BUMCA, (2) age > 20 years old, (3) singleton pregnancy, and (4) residency within the Ashkelon sub-district. Cases of documented chronic diseases or treatment with medications that may interfere with thyroid function were excluded. All in all, 202 pregnant women participated in the study. A detailed description of the screening procedure, study sample, and follow-up flow chart can be found elsewhere [16]. The BUMCA’s ethical committee approved the study (no’ 001-17-BRZ dated 7 August 2017). Following a detailed explanation of the research protocol, all participants provided written informed consent. Experienced registered dietitians conducted all dietary interviews.

### 2.2. Data Collection and Considerations

Obstetrics data, such as in vitro fertilization (IVF), parity, gravidity, treatments, gestation week, anthropometrics, GCT results, and non-fasting glucose values at delivery admission, were collected from participants’ medical files. Any GCT result above 140 or 200 mg/dL was considered abnormal or a case of GDM [17]. Any initial diabetes diagnosed during pregnancy was considered as GDM. Recorded abnormal GCT result or GDM diagnosis were considered maternal dysglycemia [17]. In addition, GDM diagnosis was made if at least two abnormal indices were observed in a 100 g Oral Glucose Tolerance Test (OGTT). The OGTT includes administering a 100 g glucose load after an overnight fast, and its indices are considered abnormal when exceedingly high plasma glucose levels are detected as follows: at the beginning of fasting (>95 mg/dL) and 1 h (>180 mg/dL), 2 h (>155 mg/dL), and 3 h (>140 mg/dL) following the OGTT [17]. Advanced maternal age was considered when delivery was performed at age > 35 years [18]. The questionnaire used in the study included self-reported information about participants’ health statuses as well as their knowledge, attitudes, and behaviors regarding nutrition and health. Smoking history was assessed using medical records and interviews. The body mass index (BMI) at delivery was calculated using data from electronic medical records based on weight and height obtained. Normal, overweight and obesity were considered when BMI was 18.5–24.9, 25.0–29.9, and above 30.0 kg/m^2^, respectively [19]. Interviews and other data collection procedures have been described in detail previously [16].

### 2.3. Assessment of Maternal Iodine Intake, Status, and Thyroid Function

This study assessed the iodine statuses of participants using three different measurements: (a) a validated, semi-quantitative iodine food frequency questionnaire (sIFFQ) used to estimate long-term iodine intake (up to a year); (b) serum thyroglobulin (Tg), which was measured to indicate intermediate-term iodine intake (weeks to months); and (c) urinary iodine concentration (UIC), which was measured as an indicator of recent iodine intake (days). The details of these measurements were previously described in full [16]. Briefly, Tg and UIC were used to describe groups’ iodine statuses, while the sIFFQ was used to reflect both each individual’s and each groups’ iodine intake [20]. In addition, the sIFFQ was administered by three registered dietitians (YSO, SRR, and YAB) who had experience working with pregnant women and who were trained in the use of the sIFFQ. The questionnaire included questions about both the initiation and duration of the administration of iodine-containing supplements (ICS) and the estimated amount of iodized salt (IS) consumed to address potential recall bias. The estimated iodine intake was classified as adequate or recommended if the sIFFQ-calculated result was equal to or greater than 160 or 220 μg/day (respectively), which are the Estimated Average Requirement (EAR) or Recommended Dietary Allowance (RDA) for iodine intake during pregnancy, respectively [21]. The RDA was used to compare the median estimated iodine intake of the studied population to the desired intake. The percentage of participants reporting inadequate iodine intake across the subgroups was determined using the EAR [21]. The UIC was determined using an inductively coupled plasma mass spectrometry method (ICP-MS), conducted according to the US CDC ICP-MS method [22]. The iodine statuses of the subgroups were considered sufficient, mild-to-moderate ID, or severe ID when the median UIC was >150, 50–150, or <50 μg/L, respectively, according to the World Health Organization (WHO) and the American Thyroid Association (ATA) guidelines [23,24].

Thyroid function was assessed using electrochemiluminescence immunoassays, as previously described [16]. Briefly, free T4 (FT4), free T3 (FT3), thyrotropin (TSH), thyroid peroxidase antibody (TPOAb), thyroglobulin antibody (TgAb), and Tg levels were determined. Reference ranges were 0.27–4.2 mU/L for TSH and 0.93–1.7 ng/dL for FT4 according to the manufacturer. Values of TgAb above 40 IU/mL and TPOAb above 35 IU/mL were considered positive, as reported elsewhere [25]. Subclinical hypothyroidism (SCH) was considered to be present when FT3 and FT4 values were normal but TSH values were above 2.5 or 4.0 mU/L during the 1st and both the 2nd and 3rd trimesters, respectively [24]. Since there has not been a consensus regarding what constitutes isolated hypothyroxinemia (IHT) or how thyroid function tests should be performed in iodine-deficient areas, reference intervals of FT4 from BUMCA and cutoffs for TSH (2.5 mU/L for the first trimester and 4.0 mU/L for both the second and third trimesters) were used to define IHT at various gestational ages [24]. According to a recently proposed population standard, a median cutoff of 13 g/L for Tg levels was considered sufficient for the entire study sample [25].

### 2.4. Neonatal Birth Data and Antheropometrics

The medical records obtained from BUMCA provided information on gender, date of birth, birth weight, length, head circumference, and Apgar score. The gestational age at birth was determined with reference to self-reports from the last menstrual period and confirmed by measuring the fetal crown–rump length. This parameter was calculated by assembling single numbers of days and the number of completed weeks. The midwives in the delivery room performed the Apgar test and measured each newborn’s weight, length, and head circumference. Birth weight percentile was calculated and standardized by employing the Israeli birth population index and using gender and gestational data, as described in detail elsewhere [26]. Cases of SGA and LGA were defined as percentiles below and above the 10th and the 90th percentiles, respectively. Low birth weight (LBW) was considered when birth weight was under 2500 g for full-term newborns. Preterm was defined as birth before 37 full weeks of gestation. Macrosomia was defined as a newborn weighing over 4000 g. Length and head circumference were calculated and standardized for the Israeli birth population based on gender and gestational age as described in detail elsewhere [27].

### 2.5. Statistical Analysis

Statistical analysis was conducted using JMP Pro software version 16 (SAS Institute, Cary, NC, USA). We used multiple linear regression models to study the association of GCT (also overlayed by severe ID vs. sufficiency assessed by UIC) and FT4 with birth weight percentile as well as estimated iodine intake (overlayed by reported ICS intake vs. none) with FT4. To test the multiple relationship between possible predictors of LGA, we used Proportional Hazards platform that fits the Cox proportional hazards model (semiparametric regression model for survival data with covariates). This model was performed by using LGA as the hazard function, gestational age as the time-to-event function, and the following 15 possible predictors as the covariates: Tg (above vs. below 13 μg/L), UIC (above vs. below 150 μg/L), parity (multiparous vs. nulliparous), ICS (any consumption during pregnancy vs. none), gravidity (multigravida vs. nulligravida), estimated iodine intake (below vs. above 220 μg/day), maternal age (advanced vs. none advanced), GDM (recorded diagnosis vs. none), GCT (abnormal vs. normal result), SCH (found vs. none), IS use (reported vs. none), delivery BMI (overweight vs. normal), FT3 (highest vs. lowest quartile), IHT (found vs. none), IVF (pre-pregnancy treated vs. none), and recruitment BMI (overweight vs. normal). In order to choose optimum split for Tg values as an LGA predictor, we used the Partition platform that recursively partitions data, creating a decision tree. The partition algorithm searched all possible splits of Tg values to best predict LGA. These splits (partitions) form a tree of decision rules. The splits continue until the desired fit is reached. The partition algorithm chooses optimum splits from many possible splits (according to the software manufacturer, JMP PRO). The categorical variables were summarized according to their amounts (percentage in brackets) and were compared by group (classified by Tg above or below the cutoff and Partition values) with the aid of the chi-square test, likelihood test, or Fisher’s exact test, as appropriate. Odds ratios (OR) were also calculated for likelihood of LGA across Tg subgroups. Abnormally distributed continuous parameters were summarized according to median and interquartile range (IQR). Continuous parameters with normal distribution were presented as mean ± standard deviation (SD). Shapiro–Wilk W or Cramer–von Mises W testing was performed to determine whether continuous parameters had a normal distribution. These parameters were compared using one-way analysis of variance (ANOVA) with Student’s t (for means), Welch’s ANOVA (for unequal variances), and the Kruskal–Wallis test (for non-parametric parameters) when appropriate. The associations between continuous GCT, FT4, and UIC values and estimated iodine intake levels were determined via linear regression (adjusting for possible confounding parameters, when necessary). A two-tailed *p* value < 0.05 was considered statistically significant.

## 3. Results

### 3.1. Study Population

Of the 202 consecutive participants screened, the results of 188 (median age, 31; age range, 20–46 years) met the eligibility criteria and were included in the study. The sociodemographic, health, obstetrics, and iodine intake features of the study population were described in detail previously [16].

### 3.2. Maternal Nutritional, Hormonal, and Clinical Characteristics

The median UIC value determined was 60 μg/L, demonstrating mild-to-moderate ID in our study population [24]. The estimated dietary iodine intake median was 179 μg/day, demonstrating lower dietary iodine intake in comparison to the RDA for pregnancy [21]. For the entire studied population, the median estimated iodine intake from ICS was 62 μg/day. The median Tg value determined was 18 μg/L; this is above the cutoff considered for iodine sufficiency in pregnancy, indicating iodine insufficiency (>13 μg/L) in this study population [25]. Detailed nutritional, hormonal, and clinical characteristics of the study population were previously described [16].

### 3.3. Maternal Characteristics, Pregnancy Outcomes, and Newborn Weight

Values of GCT correlated significantly with non-fasting glucose at delivery admission, but this correlation was found to be non-significant after multivariate analysis, adjusting for maternal age, BMI, gravidity, and parity. However, both univariate and multivariate regression analyses adjusting for age, smoking status, BMI at delivery, parity, and gravidity revealed positive correlations between maternal GCT values and adjusted newborn weight percentiles (Figure 1A). According to Figure 1B, this significant positive correlation also occurred among participants with UIC values below 150 μg/L (presumably indicating severe ID). However, this correlation did not appear among participants with UIC values above 150 μg/L (indicating a presumably sufficient iodine status), as shown in Figure 1B.

Based on the Cox model comprising 15 variables with a known or probable association with neonatal weight, maternal Tg values were the only significant predictor of LGA (adjusted hazard ratio = 3.4, 95% CI: 1.4–10.2, *p* = 0.001). As shown in Figure 2, other variables were not independently associated with LGA. In addition, the proportion of LGA was significantly higher among the newborns of participants with Tg values > 13 μg/L vs. <13 μg/L (13% vs. 4%, respectively, Table 1). The Partition algorithm chose a maternal Tg value of 17 μg/L for optimum splits to best predict LGA at birth. Accordingly, the study population was splited into two subgroups, namely, groups with Tg values above and below 17 μg/L (Table 2). The Tg > 17 μg/L subgroup exhibited significantly lower estimated iodine intake and lower ICS intake. Additionally, gravidity, parity, and LGA prevalence were significantly higher in this subgroup. Other maternal and newborn characteristics did not differ significantly across subgroups. Detailed comparisons are provided in Table 2. In addition, participants with estimated iodine intakes below the EAR were about two times more likely to have Tg > 17 μg/L (OR = 1.9 [CI 1.1, 3.5]). Furthermore, participants with Tg > 17 μg/L were five times more likely to give birth to an LGA newborn (OR = 5 [95% CI 1, 18]; *p* < 0.01, Fisher exact test).

Estimated iodine intake levels correlated differently with maternal FT4 values. The estimated iodine intake levels among the participants reporting no ICS intake were significantly and inversely correlated with maternal FT4 values, but significance did not persist following adjustments for age, smoking status, BMI at delivery, parity, and gravidity, as reported in detail in Figure 3A. On the other hand, the estimated iodine intake levels among the participants reporting ICS intake were significantly and positively correlated with maternal FT4 values that remained significant after adjustment for age, smoking, BMI, parity, and gravidity (Figure 3A). As shown in Figure 3B, the maternal FT4 values negatively correlated with birthweight percentiles. This correlation remained significant after adjustment for age, smoking, BMI, parity, and gravidity.

## 4. Discussion

This was a longitudinal prospective cohort study testing the association of both glucose and iodine intake levels with birthweight among pregnant women. Although the GCT values were correlated with adjusted newborn weight percentiles (Figure 1A), we revealed a strong independent association between a lower iodine status and increased birthweight percentiles and LGA (severe ID determined according to UIC, Figure 1B, and suggestive of iodine insufficiency, Figure 2, respectively). Additionally, estimated iodine intake was positively correlated with serum FT4 levels among the participants reporting ICS intake (Figure 3A). Moreover, adjusted maternal FT4 values were inversely correlated with newborn weight percentiles (Figure 3B). These findings may indicate that a sufficient iodine status and ICS intake constrain, and perhaps modify, the impact of maternal hyperglycemia on offspring weight. Such evidence can be crucial, as LGA newborns have a higher risk of suffering from neonatal metabolic abnormalities, including hyperglycemia, birth trauma, stillbirth, and neonatal death, and overweight, obesity, cardiovascular disease, and diabetes later in life [5,6].

Along with the possibility that iodine intake can modify the effect of hyperglycemia on birthweight, we observed that a sufficient maternal iodine status, determined with respect to a UIC > 150 μg/L, was not associated with increased birth weight, even among participants with maternal dysglycemia (Figure 1B). This observation is not in line with a recent meta-analysis conducted on 23 cohorts with 42,269 participants [28]. In that meta-analysis, birth weight was similar between groups with UIC ≥ 150 μg/L and <150 μg/L, with no evidence of linear trends [28]. The difference in the findings between that meta-analysis and our study may stem from the thorough adjustments for confounding performed in the current study and the differences in urine sample collection and exposure definitions. Additionally, our study used birthweight percentiles that were standardized for the Israeli population, while some studies included in the cited meta-analysis used z-scores to determine weight percentiles [26,28]. Moreover, the meta-analysis did not include findings regarding iodine status stratified (via UIC) into sufficient (>150 μg/L) vs. severe ID (<50 μg/L) groups [23,24], whereas such findings were included in the current study (Figure 1B). 

Moreover, our results are in line with a recent single-center study from China showing nearly significant (*p* = 0.07) levels of higher birthweights among the offspring of mothers with ID (*n* = 726) [15]. In our study population, an overall mild-to-moderate degree of ID was detected. Hence, it can be suggested that the impact of maternal dysglycemia on birthweight may also be differentiated by ID severity.

The trend shown in Figure 3 and the full mechanism behind a possible ID–FT4–LGA axis in humans are not fully established [29]. While the patterns underlying this association may be complex and need clarification, it is likely that ID leads to low FT4 levels (as iodine is an essential substrate for T4 synthesis) [30,31]. In turn, it has been well established that low thyroid hormone levels underlie insulin resistance [32,33,34], which is characterized by increased insulin secretion, in both GDM and type 2 diabetes [35,36]. Furthermore, maternal insulin can act as a growth factor in fetuses, which might lead to LGA [37,38]. On a molecular level, thyroid hormone deficiency in fetuses has been implicated with decreased lipolysis in adipocytes and decreased de novo lipogenesis in the liver via Sterol-regulatory-element-binding proteins-1 and 2 (SREBP1 and −2) [39]. Moreover, T3 has been shown to regulate Carbohydrate response element-binding protein (ChREBP) in the liver, brown adipose tissue, and the pancreas [40,41,42]. Also, T3 is a key regulator of de novo lipogenesis in various metabolic tissues [40,41]. Therefore, the determination of a possible molecular mechanism of how ID during pregnancy can influence in utero growth and development (by thyroid function) in the fetus can be constructive but remains to be elucidated.

Although we found an association between estimated iodine intake (among participants reporting ICS intake) and FT4 (Figure 3A), no cause-and-effect relationship was established. Among Asian populations, an interventional study found that ICS intake did not improve maternal thyroid function among pregnant women with mild-to-moderate ID [43]. According to our interpretation of the current study, we believe iodine status is the cause, while thyroid hormone is the effect, as supported by another interventional trial conducted among pregnant Norwegian women with mild-to-moderate ID [44]. As the current study did not explore the iron statuses of the participants or the iron content of the self-reported ICS, we cannot exclude the possibility that supplementary iron contributed to FT4. It has been shown in recent studies involving Chinese pregnant women that FT4 levels can change with iron status during pregnancy and that maternal anemia at delivery is a risk factor for low birth weight [45,46]. Therefore, it is important to investigate the effect of both iron and iodine intake on maternal thyroid function and the weight of offspring. 

The relatively low proportion of participants presenting a sufficient iodine status (Tg < 13 μg/L, 68%, Table 1) [25] along with the high proportion of participants reporting an inadequate iodine intake below the EAR (160 μg/day, 41%, Table 2) [20] and levels that might correspond to the iodine status of cases of severe ID according to UIC (<50 μg/L, Table 2) [23,24] should all raise concerns for public health. Two major reasons may explain this ongoing hazardous condition: (1) Israel lacks a mandatory iodine fortification program [16,23], and only nine participants (only 4% of the total study population) reported voluntary consumption of iodized table salt; (2) despite Ministry of Health’s guidelines recommending prenatal ICS use at least one month prior to conception, only 6% of the current study participants reported pre-pregnancy initiation of ICS intake. Moreover, the estimated iodine intake from ICS was 62 μg/day, although the current study participants reported that the majority of ICS they consumed contained 220 μg of iodine per tablet. The reason for this gap is the inconsistent ICS intake values reported by the ICS consumers who participated in our study, resulting in a weighted lower estimate of iodine intake from ICS (Table 2) [16]. In light of this, safe and mandatory iodine prophylaxis in Israel should be urgently implemented. Moreover, the evidence shown in the current study could highlight the ‘real-life’ vital need to efficiently increase the prescription rate of ICS pre-conceptionally.

Our study has several notable strengths. The cohort analyzed was of sufficient size for monitoring iodine status in the studied population. The parallel quantification of three iodine status biomarkers (sIFFQ, Tg, and UIC) allowed for a detailed assessment of the patients’ iodine statuses and provided a comprehensive picture of the prevalence of ID among the participants. In addition, it contributed to a deeper understanding of our hypothesis that iodine intake may modify the effect of maternal dysglycemia on excessive fetal growth. 

Our study also had several limitations. One such limitation was the use of spot UIC to estimate subgroups. Of note, the minimum number of spot urine samples needed to estimate populational iodine statuses within a precision range of ±10% is about 125 participants [47]. However, the same pattern in relation to the GCT–ID–LGA axis was also observed using the sIFFQ (Figure 3A) and Tg (Figure 1 and Table 2). In addition, we did not determine iodine status (by either UIC or Tg) during the first trimester, which is a critical timeframe for fetal development [9,23]; thus, we cannot rule out a link between early gestational ID and LGA. Nevertheless, as the timeframe of the sIFFQ was 1 year and we analyzed the weighted estimate of iodine intake through ICS administration and its initiation, we can assume that pre-pregnancy ID was probably present in the current study population due to the lower contribution of ICS along with the low proportion of iodized salt use (Table 2) [16]. Another potential limitation was the absence of data regarding maternal iron statuses throughout pregnancy, which could affect both thyroid function and fetal growth [35]. Moreover, we did not measure the estimated fetal weight percentile during pregnancy. Thus, the reduced maternal iron status and estimated fetal weight gain during pregnancy observed might weaken the hypothesis regarding a human ID–FT4–LGA axis. We plan to explore this hypothesis in future work.

## 5. Conclusions

This study suggests that an insufficient iodine status during pregnancy might increase the risk of LGA in mild-to-moderate ID regions. Sufficient iodine statuses and ICS intake levels may modify and constrain the effect of maternal hyperglycemia on offspring overgrowth. Further investigations should focus on using maternal T4 to inspect the relationship between maternal ID and newborn weight among pregnant women with maternal dysglycemia.

## Figures and Tables

**Figure 1 nutrients-15-02914-f001:**
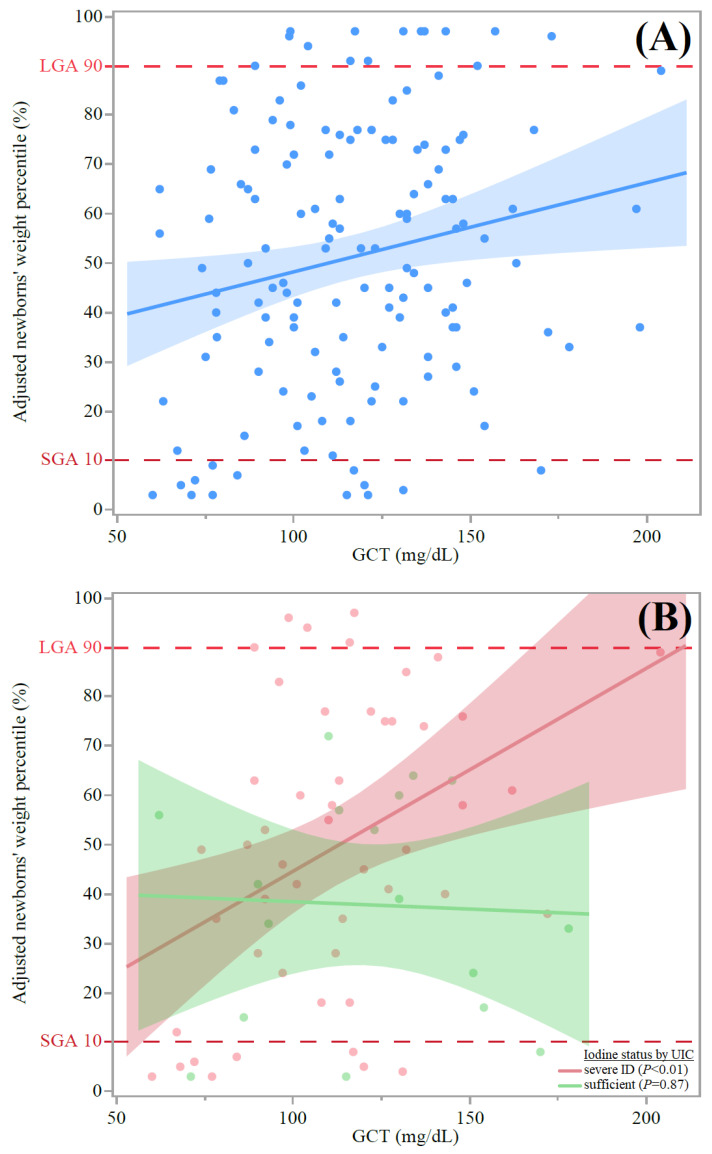
Demonstration of the correlation between maternal GCT and newborns’ birth percentiles. (**A**) Scatter plot of adjusted newborn weight percentiles (*y* axis) in comparison to GCT values (*n* = 171) (*x* axis) of all participants with available GCT results (*n* = 171) (where dashed vertical lines show LGA and SGA): y = 30.2 + 0.2(x), and R^2^ = 0.039; β (95% CI) = 0.20 (0.03, 0.33), *p* = 0.018. The association remained significant according to multivariate regression analysis adjusting for the ages of the pregnant women, smoking status, BMI at delivery, parity, and gravidity: β (95% CI) = −0.21 (0.03, 0.36), *p* = 0.022. (**B**) Scatter plot of newborn weight percentiles (*y* axis) adjusted by GCT values (*n* = 86), excluding participants with mild-to-moderate ID (*x* axis), with lines to show linear fit (dashed vertical lines show LGA and SGA). For participants with sufficient iodine status (according to UIC): y= 41.4 + 0.03(x), R^2^ = 0, *p* = NS. For participants with severe ID (according to UIC): y = 3.5 + 0.4(x), and R^2^ = 0.16, β (95% CI) = 0.33 (0.09, 0.55), *p* < 0.01. The association remained significant following multivariate regression analysis adjusting for the ages of the pregnant women, smoking status, BMI at delivery, parity, and gravidity: β (95% CI) = 0.4 (0.37, 1.62), *p* < 0.01. Abbreviations: GCT, glucose challenge test; LGA, large for gestational age; SGA, small for gestational age; CI, confidence intervals; ID, iodine deficiency; UIC, urinary iodine concentrations; BMI, body mass index.

**Figure 2 nutrients-15-02914-f002:**
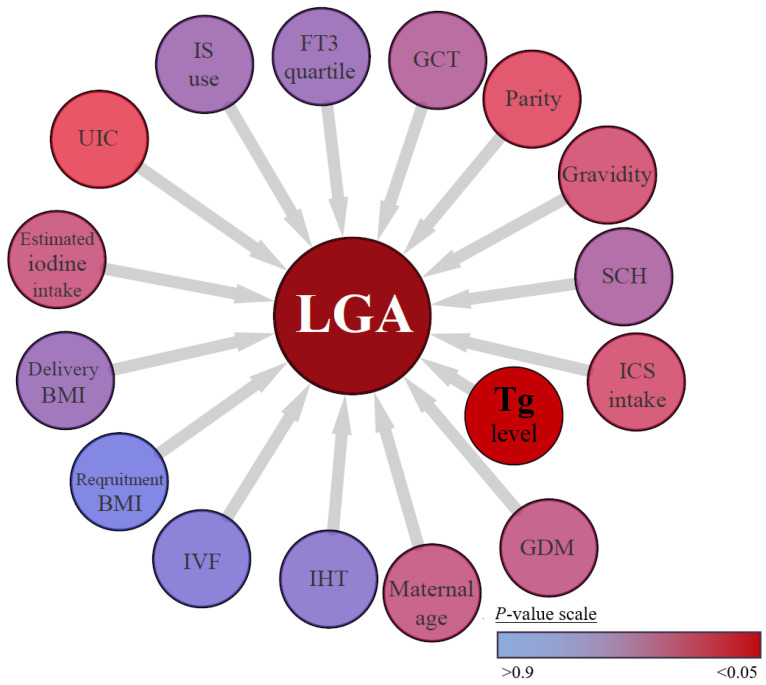
Multiple maternal variables that possibly influence LGA risk according to Proportional Hazards model. Tg > 13 g/L was independently associated with LGA (adjusted hazard ratio = 3.4, 95% CI: 1.4–10.2, *p* = 0.001, Cox proportional hazards model; time for event—total gestational age at birth), while all other variables were not. LGA, large for gestational age; GDM, gestational diabetes mellitus; IHT, isolated hypothyroxinemia (FT4 < 0.93 ng/L, TSH < 2.5 mU/L, or 4.0 mU/L during 1st and both 2nd and 3rd trimesters, respectively); IVF, In vitro fertilization; BMI, body mass index; UIC, urinary iodine concentration; IS, iodized salt; FT3, free triiodothyronine; GCT, glucose challenge test; SCH, subclinical hypothyroidism (TSH > 2.5 mU/L or 4.0 mU/L during 1st and both 2nd and 3rd trimesters, respectively, along with normal FT3 and FT4 values); ICS, iodine-containing supplement.

**Figure 3 nutrients-15-02914-f003:**
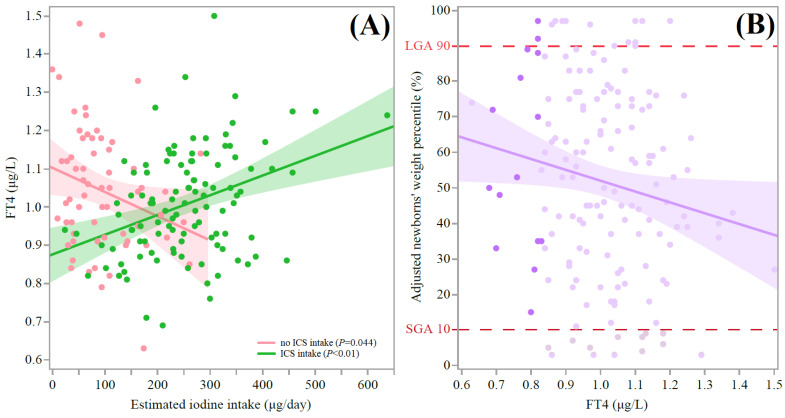
Overview of the possible association between maternal estimated iodine intake and newborn birth percentiles. (**A**) Scatter plot of FT4 values (*y* axis) with respect to estimated iodine intake levels (*x* axis) divided (overlay) by self-reports on ICS intake at any period of pregnancy (*n* = 178). For participants reporting no ICS intake throughout gestation, y = 1.1 − 0.001(x), R^2^ = 0.06, β (95% CI) = −0.25 (0, −0.0001), *p* = 0.044. The association did not remain significant after multivariate regression analysis adjusting for the ages of pregnant women, smoking status, BMI at delivery, parity, and gravidity. For participants reporting ICS intake at any time-period of pregnancy: y = 0.9 – 0.001(x), R^2^ = 0.13, β (95% CI) = 0.36 (0.0003, 0.0008), *p* < 0.001. The association remained significant in multivariate regression analysis adjusting for the ages of pregnant women, smoking status, BMI at delivery, parity, and gravidity: β (95% CI) = 0.35 (0.0002, 0.0008), *p* = 0.001. (**B**) Scatter plot of adjusted newborn weight percentiles (*y* axis) with respect to FT4 values (*n* = 173) and excluding participants with GDM (*x* axis), where dashed vertical lines show LGA and SGA (dark-purple points represent participants with IHT): y = 82.6 − 30.1(x), R^2^ = 0.038, β (95% CI) = −0.17 (−1.64, −59.52), *p* = 0.024. The association remained significant in multivariate regression analysis adjusting for the ages of pregnant women, smoking status, BMI at delivery, parity, and gravidity: β (95% CI) = −0.15 (−0.08, −56.49), *p* = 0.049. The solid lines represent the estimated linear fit, and the shaded areas illustrate the 95% CIs. Abbreviations: FT4 = free thyroxine, ICS = iodine-containing supplement, LGA = large for gestational age, SGA = small for gestational age, IHT = isolated hypothyroxinemia, and CI = confidence interval.

**Table 1 nutrients-15-02914-t001:** Selected maternal characteristics and their offspring’s selected birthweight indices.

Subgroups Segregated by Maternal Tg Values	>13 μg/L	≤13 μg/L	*p* Value
**Pregnant women, n**	127	61	
Gestational age (weeks) at recruitment, mean ± SD	31 ± 1	31 ± 2	NS
GCT (mg/dL), mean ± SD	118 ± 28	113 ± 31	NS
BMI (kg/m^2^)			
At recruitment, mean ± SD	28 ± 5	28 ± 5	NS
At delivery, mean ± SD	30 ± 6	29 ± 4	NS
Iodine Intake			
Estimated dietary Iodine intake (μg/d), mean ± SD *	181 ± 109	215 ± 118	0.06
Iodized salt use, n (%) ^ם^	4 (5)	5 (11)	NS
ICS intake, n (%) *^F^*	68 (58)	38 (69)	NS
Birthweight			
Adjusted weight percentile (%), mean ± SD *^Do^*	53 ± 28	51 ± 25	NS
LGA *^L^*	14 (13)	4 (4)	0.04

Tg, thyroglobulin; SD, standard deviation; NS, not significant; GCT, 50 g 1 h post-oral glucose challenge test; BMI, body mass index; ICS, iodine-containing supplement; LGA, large for gestational age. * Significant difference (Student’s *t* test, α = 0.05); ^ם^, iodized salt, 3 μg iodine/100 gr; *^L^* significant difference (Likelihood ratio test, α = 0.05); *^Do^* adjusted for gestational age and gender according to Israeli birth weight standards [26].

**Table 2 nutrients-15-02914-t002:** Sociodemographic, anthropometric, and clinical characteristics of pregnant women and their offspring.

Subgroups Segregated by Maternal Tg Values	>17 μg/L	≤17 μg/L	*p* Value
**Pregnant women, n**	96	92	
Age (y), mean ± SD	31 ± 6	32 ± 5	NS
Gestational age (weeks) at recruitment, mean ± SD	32 ± 7	31 ± 7	NS
Israeli born, n (%)	52 (54)	42 (46)	NS
Tertiary education, n (%)	42 (44)	52 (56)	NS
Secular n (%)	28 (30)	35 (38)	NS
IVF	6 (6)	10 (11)	NS
Smoking			
Current smoker	13 (14)	11 (12)	NS
Past smoker	18 (19)	18 (20)	NS
Alcohol, n (%)	0 (0)	0 (0)	NS
Post-psychological stressful event, n (%)	14 (15)	18 (20)	NS
GCT (mg/dL), mean ± SD	119 ± 28	114 ± 31	NS
BMI (kg/m^2^)			
At recruitment, mean ± SD	29 ± 5	28 ± 5	NS
At delivery, mean ± SD	31 ± 6	29 ± 4	NS
Gravidity, mean ± SD *	4 ± 2	3 ± 2	0.05
Parity, mean ± SD *	3 ± 2	2 ± 1	0.03
Iodine Intake			
Estimated dietary Iodine intake (μg/d), mean ± SD *	163 ± 104	221 ± 114	<0.01
Iodine intake < EAR, n (%) *^F^*	47 (49)	30 (33)	0.03
Iodized salt use, n (%) ^ם^	4 (4)	5 (5)	NS
ICS intake, n (%) *^F^*	44 (46)	62 (67)	<0.01
Estimated iodine intake from ICS (μg/d), median (IQR) *^K^*	1 (0–150)	150 (0–220)	<0.01
Dietary goitrogens exposure, n (%) *^F^*	20 (22)	12 (13)	NS
UIC			
Median UIC, μg/L (IQR)	53 (39–86)	65 (41–97)	NS
Participants with UIC <150 μg/L, n (%)	77 (80)	72 (86)	NS
Participants with UIC <50 μg/L, n (%)	34 (35)	30 (33)	NS
TSH			
Mean ± SD (mIU/L), n (%)	1.8 ± 1.0	1.8 ± 1.0	NS
Participants with SCH, n (%)	4 (4)	2 (2)	NS
FT4			
Mean ± SD (μg/L), n (%)	1.0 ± 0.2	1.0 ± 0.1	NS
Participants with IHT, n (%)	7 (7)	8 (9)	NS
FT3 (pmol/L), mean ± SD	4.1 ± 0.7	3.9 ± 0.7	NS
TPO Ab			
TPO Ab (mIU/L), median (IQR)	13 (11–16)	13 (11–17)	NS
Positive TPO Ab, n (%)	1 (1)	4 (4)	NS
Tg Ab			
Tg Ab (mIU/L), median (IQR)	10 (10–11)	10 (10–12)	NS
Positive Tg Ab, n (%)	1 (1)	4 (5)	NS
**Newborns at birth ^a^, n**	85	84	
Gestational age (days), mean ± SD	266 ± 29	270 ± 13	NS
Preterm birth, n (%)	13 (15)	11 (13)	NS
Gender (Female, Male)	36, 47	36, 48	NS
Apgar score			
At 1 min after delivery, mean ± SD	8.9 ± 0.1	8.9 ± 0.1	NS
At 5 min after delivery, mean ± SD	9.9 ± 0.4	9.9 ± 0.3	NS
Birthweight			
Crude weight (g), mean ± SD	3176 ± 652	3029 ± 580	NS
LBW	3 (4)	8 (10)	NS
Macrosomia	6 (7)	2 (2)	NS
Adjusted weight percentile (%), mean ± SD *^Do^*	56 ± 28	49 ± 26	0.07
SGA	5 (6)	10 (12)	NS
LGA *^F^*	13 (15)	3 (3)	0.02
Length percentile (%), mean ± SD *^Da^*	74.4 ± 26.3	70.6 ± 25.9	NS
Head circumference (cm)			
Mean ± SD	34.4 ± 2.1	34.0 ± 1.7	NS
> 90th percentile *^Da^*, n (%)	27 (28)	17 (18)	NS

Tg, thyroglobulin; SD, standard deviation; NS, not significant; IVF, in vitro fertilization; GCT, 50 g 1 h post-oral glucose challenge test; BMI, body mass index; EAR, estimated average requirements (160 μg/d); ICS, iodine-containing supplement; IQR, interquartile range; UIC, urinary iodine concentration; TSH, thyroid-stimulating hormone (i.e., thyrotropin); SCH, subclinical hypothyroidism (TSH < 2.5 mU/L or 4.0 mU/L during 1st and both 2nd and 3rd trimesters); FT4, free thyroxine; IHT, isolated hypothyroxinemia (FT4 < 0.93 ng/L, TSH > 2.5 mU/L or 4.0 mU/L during 1st and both 2nd and 3rd trimesters); FT3, Free triiodothyronine; TPO Ab, thyroid peroxidase antibodies; Tg Ab, thyroglobulin antibodies; LBW, birth weight below 2500 g among full-term newborns; SGA, small for gestational age; LGA, large for gestational age. * Significant difference (Student’s *t* test, α = 0.05); ^ם^ iodized salt, 3 μg iodine/100 gr; *^F^* significant difference (Fisher’s Exact test, α = 0.05); *^K^* significant difference (Kruskal–Wallis test, α = 0.05); ^a^ For only newborns with available data; *^Do^* Adjusted for gestational age and gender according to Israeli birth weight standards [26]; *^Da^* Adjusted for birth week and gender according to birth length and head circumference Israeli standards.

## Data Availability

Data described in the manuscript, code book, and analytic code will be made available upon request following application and approval.

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
