# Peer review of "Does Iodine Intake Modify the Effect of Maternal Dysglycemia on Birth Weight in Mild-to-Moderate Iodine-Deficient Populations? A Mother–Newborn Prospective Cohort Study"

_nutrients, 2023, doi:10.3390/nu15132914_

Round 1
Reviewer 1 Report
How maternal glycemic status and maternal iodine status influence the birth weight is a unclear problem for the research of iodine deficiency. Through a cohort study, it illustrated the associations between birth weight and both maternal glucose levels and iodine intake in pregnant women with mild-to-moderate ID. The analysis of this paper is comprehensive and interesting, however, there are several issues to be addressed before it published.
The main problems are as follows:
1. It is known that urine iodine concentration is suitable for the evaluation of iodine nutrition in population, however, in individual, it is not a good indicator. The dietary iodine intake could reflect the status of individual iodine nutrition. In this study, both urine iodine concentration and iodine dietary intake were investigated. Why not use the latter one to evaluate and analyze in this paper?
2. It is interesting that the affect of dysglycemia on birth weight were inconsistent in pregnant women with UIC <150 μg/L and UIC >150 μg/L. How to explain it? What are the possible mechanisms?
3. As the cutoff ≤13 μg/L for Tg levels were used in the cox model, and it is suggested that the details need to be described by Tg above and below 13 μg/L in Table 1, although the decision tree chose maternal Tg value of 17 μg/L for optimum splits to best predict LGA at birth.
4. The color in Figure 2 is unclear, and it is difficult to recognize the statistically significant variables.
Author Response
Dear Editor,
We appreciate the positive and constructive review. All comments and corrections were incorporated in the current version of the manuscript. A point-by-point response to the comments of Reviewer 1 follows below.
Thank you for your consideration.
Yaniv Ovadia
Reviewer #1:
How maternal glycemic status and maternal iodine status influence the birth weight is a unclear problem for the research of iodine deficiency. Through a cohort study, it illustrated the associations between birth weight and both maternal glucose levels and iodine intake in pregnant women with mild-to-moderate ID. The analysis of this paper is comprehensive and interesting, however, there are several issues to be addressed before it published.
The main problems are as follows:
- It is known that urine iodine concentration is suitable for the evaluation of iodine nutrition in population, however, in individual, it is not a good indicator. The dietary iodine intake could reflect the status of individual iodine nutrition. In this study, both urine iodine concentration and iodine dietary intake were investigated. Why not use the latter one to evaluate and analyze in this paper?
Thank you for for this helpful comment. Clarifications were performed and interpretation is now more cautious (lines 30-32, 129-131, 139,140).
- It is interesting that the affect of dysglycemia on birth weight were inconsistent in pregnant women with UIC <150 μg/L and UIC >150 μg/L. How to explain it? What are the possible mechanisms?
Thank you. Explanation is now provided (lines 347-362). A possible alternative mechanism is also suggested (lines 364-370).
- As the cutoff ≤13 μg/L for Tg levels were used in the cox model, and it is suggested that the details need to be described by Tg above and below 13 μg/L in Table 1, although the decision tree chose maternal Tg value of 17 μg/L for optimum splits to best predict LGA at birth.
Thank you. Relevant table added (lines 280-286).
- The color in Figure 2 is unclear, and it is difficult to recognize the statistically significant variables.
Thank you. The color of the only significant variable was strengthened (line 268).

Reviewer 2 Report
This original paper assesses the effect of iodine deficiency on maternal glucose metabolism during late pregnancy and, consequently, on neonatal outcomes such as neonatal weights (LGA) in a cohort of 202 Israeli pregnant women. The study could be of interest, as also discussed by authors, since mild or moderate iodine deficiency may also deteriorate insulin sensitivity, contributing as an independent risk factor of poor neonatal outcomes.
Nevertheless, the authors should address some issues, and the paper requires to be adjusted accordingly.
1) The title needs to better represent the study content. As some examples, it should include explicit information on the study findings and study design.
2) The study population should be described in more detail in the methods section. How was the patients' eligibility evaluated? What were the inclusion and exclusion criteria? It could be useful to write the number of the global population of pregnant women explicitly and then provide the percentage of those included in the analyses.
3) The 50-g glucose challenge test is an acceptable screening test to diagnose GDM. However, it should be considered that a negative test (i.e., glucose <140 mg/dL) does not entirely role out the diagnosis, and an OGTT is also compulsory to exclude any glucose imbalance in pregnancy. How do the authors explain this issue?
4) It is just an observation. Iodine prophylaxis is recommended before and during pregnancy to prevent iodine deficiency, even subclinical, as iodine-related disorders are particularly relevant in the first trimester of pregnancy. However, it is unclear why the number of pregnant women achieving sufficient iodine exposure was so small.
5) It is just an observation. Chronic iodine consumption (1-year exposure), assessed by the IFFQ, indicated that a mild-to-moderate iodine deficiency was already evident before the pregnancy in these 202 pregnant women. Hence, the data may be another characteristic trait of insufficient iodine exposure. Have the authors considered data from the IFFQ to implement the analyses better to estimate the iodine deficiency role on the LGA outcome? Please briefly discuss it.
6) Why have the authors considered the LGA as the only negative outcome of pregnancy? Please provide some explanation in the discussion section.
1) Please check the text for possible typing errors.
Author Response
Dear Editor,
We appreciate the positive and constructive review. All comments and corrections were incorporated in the current version of the manuscript. A point-by-point response to the comments of Reviewer 2 follows below.
Thank you for your consideration.
Yaniv Ovadia
Reviewer #2:
This original paper assesses the effect of iodine deficiency on maternal glucose metabolism during late pregnancy and, consequently, on neonatal outcomes such as neonatal weights (LGA) in a cohort of 202 Israeli pregnant women. The study could be of interest, as also discussed by authors, since mild or moderate iodine deficiency may also deteriorate insulin sensitivity, contributing as an independent risk factor of poor neonatal outcomes.
Nevertheless, the authors should address some issues, and the paper requires to be adjusted accordingly.
1) The title needs to better represent the study content. As some examples, it should include explicit information on the study findings and study design.
Thank you for drawing our attention. Title is revised now (lines 2-5).
2) The study population should be described in more detail in the methods section. How was the patients' eligibility evaluated? What were the inclusion and exclusion criteria? It could be useful to write the number of the global population of pregnant women explicitly and then provide the percentage of those included in the analyses.
Thank you. Detailed description is now included within the Methods section (lines 91-98).
3) The 50-g glucose challenge test is an acceptable screening test to diagnose GDM. However, it should be considered that a negative test (i.e., glucose <140 mg/dL) does not entirely role out the diagnosis, and an OGTT is also compulsory to exclude any glucose imbalance in pregnancy. How do the authors explain this issue?
Thank you. The issue is now addressed within the Methods section (lines 110-114).
4) It is just an observation. Iodine prophylaxis is recommended before and during pregnancy to prevent iodine deficiency, even subclinical, as iodine-related disorders are particularly relevant in the first trimester of pregnancy. However, it is unclear why the number of pregnant women achieving sufficient iodine exposure was so small.
Thank you. The issue is now discussed in detail within Discussion (lines 393-408).
5) It is just an observation. Chronic iodine consumption (1-year exposure), assessed by the IFFQ, indicated that a mild-to-moderate iodine deficiency was already evident before the pregnancy in these 202 pregnant women. Hence, the data may be another characteristic trait of insufficient iodine exposure. Have the authors considered data from the IFFQ to implement the analyses better to estimate the iodine deficiency role on the LGA outcome? Please briefly discuss it.
Thank you. The issue is now discussed in detail within Discussion (lines 398-405, 419-425).
6) Why have the authors considered the LGA as the only negative outcome of pregnancy? Please provide some explanation in the discussion section.
Thank you. The issue is now explained Discussion (lines 343-346).

Round 2
Reviewer 2 Report
Dear EiC, and Dear authors, the paper has been adjusted significantly after the first round of revision according to my point of view.
As a recommendation, I suggest including the term "study" in the title, i.e., "A mother-newborn prospective cohort study". Please note that an additional review step is no longer required to approve this minimal change.
Given that iodine fortification programs are not mandatory in your country, the works could highlight the real-life need to increase the prescription rate of iodine-containing nutraceuticals pre-conceptionally and during pregnancy and lactation.
Author Response
We appreciate the positive and constructive review. Comments and corrections were incorporated in the current version of the manuscript. A point-by-point response to the comments of the reviewer follows below.
1) Regarding including the term "study" - The title is now revised accordingly (line 4).
2) Regarding highlighting the real-life need to increase the prescription rate of ICS - The Discussion is now revised accordingly (lines 415-417).
Thank you for your consideration.
Yaniv S. Ovadia, PhD RD